

# Community based actions save Yellow-spotted river turtle (*Podocnemis unifilis*) eggs and hatchlings flooded by rapid river level rises

Darren Norris[1,2,3], Fernanda Michalski[1,2,4] and James P. Gibbs[5]

[1] Ecology and Conservation of Amazonian Vertebrates Research Group, Federal University of Amapá, Macapá, Amapá, Brazil
[2] Postgraduate Programme in Tropical Biodiversity, Federal University of Amapá, Macapá, Amapá, Brazil
[3] School of Environmental Sciences, Federal University of Amapá, Macapá, Amapá, Brazil
[4] Pro-Carnivores Institute, Atibaia, São Paulo, Brazil
[5] Department of Environmental and Forest Biology, State University of New York, Syracuse, NY, USA

Corresponding author
Darren Norris,
darren.norris@unifap.br

## ABSTRACT

The conservation and recovery of increasingly threatened tropical freshwater turtle populations depends on effective management plans and actions. Here we show that community-based actions saved Yellow-spotted river turtle (*Podocnemis unifilis*) eggs submerged by unseasonal flooding and ensured the release of hatchlings. We recovered 926 eggs and 65 premature hatchlings from 74 submerged nests at 16 flooded nesting areas along 75 km of waterways. The rescued eggs were transferred to a rearing center and incubated. Hatchlings emerged from eggs that had remained underwater for up to two days. Hatchlings were maintained in 250–500 L nursery tanks until yolk sac scars had closed. Healthy hatchlings were then immediately released around the original nesting areas. We released 599 hatchlings (60.4%) from 991 submerged eggs and hatchlings. Egg survival (61.7% (571/926)) was substantially less than hatchling survival (94.2% (599/636)) but within the expected range of values reported for this species. These findings suggest that Yellow-spotted river turtle eggs and embryos are resistant to short-term submersion, which could help explain the widespread distribution of this species across highly seasonal Amazonian rivers. Management plans should take the possible survival of submerged eggs into consideration as part of species conservation and recovery actions.

## INTRODUCTION

Like many tropical species Amazonian freshwater turtles are threatened by deforestation (*Fagundes et al., 2018*), climate change (*Eisemberg et al., 2016*), and unsustainable exploitation (*Rachmansah, Norris & Gibbs, 2020*; *Smith, 1979*). Precautionary estimates suggest that populations of the once abundant Yellow-spotted river turtle (*Podocnemis*

*unifilis*) may experience severe (≥50%) and rapid (<50 years) future losses across 60% (5.3 M km$^2$) of the pan-Amazonian range (*Norris et al., 2019*). The conservation and recovery of this and other Amazonian freshwater turtles will therefore depend on effective and active management plans that are likely to be more successful with local community involvement (*Campos-Silva et al., 2018*; *Harju, Sirén & Salo, 2018*; *Norris, Michalski & Gibbs, 2018b*; *Norris et al., 2019*).

Increasingly frequent alterations in the seasonal Amazon flood pulse may seriously impact the region's flora and fauna (*Barichivich et al., 2018*; *Marengo & Espinoza, 2016*). Unpredictable water level rises are known to strongly affect freshwater turtle recruitment along seasonally flooded rivers (*Bodie, 2001*; *Semlitsch & Bodie, 2003*; *Steen et al., 2012*). Whilst some turtles show adaptations to predictable changes in water levels (*Kennett, Christian & Pritchard, 1993*), extreme flooding events cause dramatic increases in egg and embryo mortality in South American Podocnemididae (*Eisemberg et al., 2016*; *Páez et al., 2015*).

Nesting in members of the Podocnemididae is usually synchronized to avoid the seasonal flood pulse in lotic waterways. However, elevated nest loss due to extreme and/or unseasonal flooding has been widely documented in *Podocnemis erythrocephala* (*Batistella & Vogt, 2008*), *P. expansa* (*Eisemberg et al., 2016*), *P. lewyana* (*Gallego-García & Castaño-Mora, 2008*), *P. sextuberculata* (*Vogt & Pezzutti, 1999*) and *P. unifilis* (*Caputo, Canestrelli & Boitani, 2005*). The impact of nest flooding is highly variable in space and time depending on difficult-to-predict factors such as the start of the rainy season and its intensity (*Eisemberg et al., 2016*). For example, flooding of *P. unifilis* nests caused annual losses of less than 10% in Brazil (*Pignati et al., 2013*), 63% in Ecuador (*Caputo, Canestrelli & Boitani, 2005*), 64% in Colombia (*Páez & Bock, 1998*) and 1% to 100% in Peru (*Soini, 1995*). Thus, although the widespread distribution of these species demonstrate an evolutionary stable adaption to fluctuation of egg and juvenile mortality due to flooding across the decades of female reproduction, rapid river level rises are a major factor affecting turtle nest mortality. Indeed, 21st century changes across Amazonian waterways (*Castello et al., 2013*) are driving increasing mortality across the species' ranges (*Lovich et al., 2018*; *Páez et al., 2015*; *Rhodin et al., 2018*).

Although the biology of and threats to *P. unifilis* are well documented, there is still limited evidence of the efficacy of different management options for conservation of *P. unifilis* (*Páez et al., 2015*). Increased population losses are expected due to widespread and abrupt alterations in river flow patterns caused by climate change, habitat loss and hydroelectric expansion (*Castello et al., 2013*; *Eisemberg et al., 2016*; *Fagundes et al., 2018*). Indeed, dams present a unique challenge to freshwater turtles in the area of reservoir formations due to irreversible submersion of nesting areas (*Alho, 2011*; *Lees et al., 2016*; *Norris, Michalski & Gibbs, 2018a*). Yet, we lack solutions to the known impacts of submersion on *P. unifilis* nest sites.

Here we present the results from community based recovery of *P. unifilis* eggs and premature hatchlings submerged by unprecedented rapid river level rises. We ask how many of the flooded eggs remain viable and if hatchlings were apparently healthy. We use

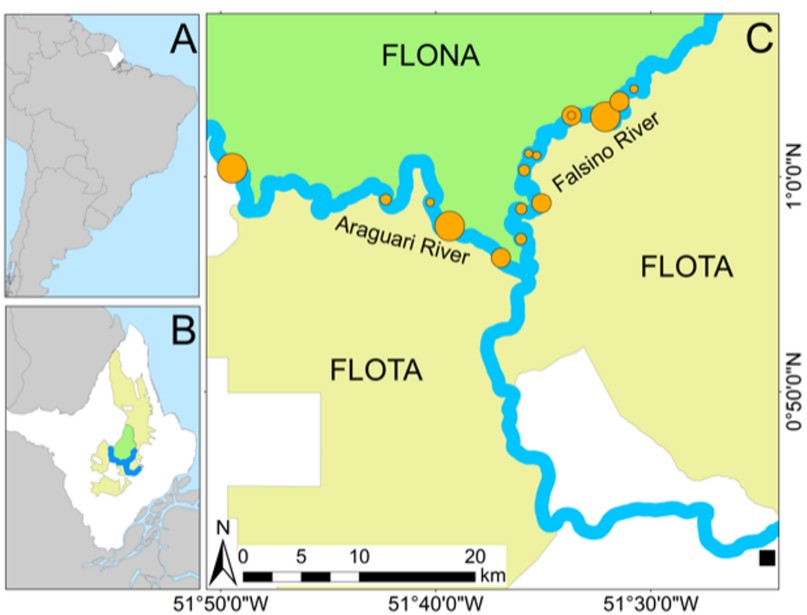

**Figure 1 Study area.** (A) State of Amapá in Brazil. (B) Location of the study area, and protected areas (FLONA and FLOTA) within Amapá. (C) Location of 16 *Podocnemis unifilis* nesting areas where 74 nests were recovered after submersion (filled orange circles sized in proportion to the number of recovered eggs). Location of the nearest town (Porto Grande) is shown by a black square.

our results to discuss how management plans could include rescue of submerged eggs and premature hatchlings as part of species conservation and recovery actions.

## METHODS

### Ethics statement

All methods were carried out in accordance with relevant guidelines and regulations. The actions we report were developed and applied following international (*Beaupre et al., 2004*) and national (*Balestra, 2016*) guidelines. Fieldwork and data collection were conducted under research permit numbers SISBIO 63668-1 and 63668-2 to DN, issued by the Brazilian Ministério do Meio Ambiente ("MMA"). Fieldwork with turtles was approved by the Animal Use Ethics Committee from the Federal University of Amapá (UNIFAP) (CEUA-UNIFAP approval 009/2017).

### Study area

The study was conducted in the Araguari river basin, located in the Brazilian State of Amapá (Fig. 1). Nesting of *Podocnemis unifilis* has been monitored since 2011 in the study area and here we present a brief summary of the previously described study area and *P. unifilis* nesting areas (*Michalski et al., 2020*; *Norris & Michalski, 2013*; *Norris, Michalski & Gibbs, 2018a, 2018b*; *Quintana et al., 2019*). Climate in the area is characterized as equatorial monsoon (*Kottek et al., 2006*) with an annual rainfall greater than 2,000 mm. Rainfall is strongly seasonal, with the dry season extending between September and November

(total monthly rainfall <150 mm), and the wet season from February to April (total monthly rainfall >300 mm (*De Oliveira, Norris & Michalski, 2015*; *Paredes et al., 2017*)).

## Data collection

As part of a collaborative monitoring scheme with the local community, we monitored 29 *P. unifilis* nesting areas from September to December 2018. This is a representative subset of the available nesting areas described previously (*Norris, Michalski & Gibbs, 2018a*; *Quintana et al., 2019*) that were chosen in 2018 to encompass broad gradients (environmental and anthropogenic) and include areas with the majority of nests (including nesting areas with >90% nests recorded in previous years). Nesting areas were visited at least once a month to search for turtle nests and to monitor conditions of nests that were individually marked with a small stick. Nests were located by following turtle tracks on the sandy/gravel substrates and systematic substrate searches with blunt wooden sticks and fingertips (*Escalona & Fa, 1998*; *Norris, Michalski & Gibbs, 2018a*; *Quintana et al., 2019*). Searches were conducted by teams of two to three observers along lengthwise transects throughout each area at a standardized speed (mean 0.8, range 0.2–1.3 km per hour); time spent searching nesting areas ranged from 10 to 97 min depending on the size of the area (*Norris, Michalski & Gibbs, 2018a*).

## Recovery of submerged eggs

In 2018, the work with local communities could be classified as collaborative monitoring with external data interpretation (*Danielsen et al., 2009*). In addition to monitoring nesting areas, members of the local communities also participated in management actions including installation and monitoring of nest predator exclusion devices (*Norris, Michalski & Gibbs, 2018b*). Previously, strategically targeted community-based actions at only two nesting areas enabled the successful protection of 75% of nests along a 33 km stretch of river (*Norris, Michalski & Gibbs, 2018b*). In 2018 actions were expanded to include two monitoring teams working simultaneously (one team on each of the Araguari and Falsino rivers).

In 2018 there was an unprecedented and rapid rise in river levels (Fig. 2). On the 29 November 2018 the field team from the Falsino river reported rising river levels due to unseasonal early rains and that on 28 November some nests had been moved to higher points on at least one nesting area. Based on our knowledge from previous years in the region there was no expectation that the river water level would continue to rise. However, after assessing the river level at the field site on 30 November it became clear that the rapid water level rise was flooding nests along both Araguari and Falsino rivers. Members of the local community living in the region for more than 30 years remember early rain, but do not recall river levels rising so quickly. Hence, as this was an unprecedented event, our team took action to try and save at least some of the nests.

To reduce the loss of eggs and hatchlings our plan of action followed these steps (Fig. 3):

1. Move nests to higher locations that would not be flooded keeping them in the same nesting area where eggs were laid.
2. Locate and collect submerged eggs and hatchlings where possible.

3. Transport and dry sand from the nesting beaches (needed for incubating eggs and strengthening of hatchlings).

4. Set up incubators for eggs.

5. Transfer hatchlings to a "nursery" for strengthening.

6. Release hatchlings in the same region where they were rescued.

By 1 December 2018 it was clear that the river levels would not decline and that we would need to take additional steps (i.e., moving nests to higher points within the same nest site was no longer a viable option). We established an improvised rearing facility located on the property of a local landowner. We created incubators for eggs and temporary nursery tanks for premature hatchlings. Eggs were placed in incubators (50–80 L polystyrene foam boxes) filled with sand from the submerged beaches and heated by lamps with the aid of a generator that was turned on at least four hours a day (Fig. 3). Hatchlings were transferred to water tanks (250 or 500 liter) that served as nursery areas for strengthening and to enable complete absorption of the yolk sac (Fig. 3). In the nursery tanks fresh vegetation (cassava (*Manihot esculenta*) leaves) was also provided for both food and shelter. Hatchlings were periodically (every 2–3 days) weighed and measured (straight-line carapace length and width) to monitor their health and development.

Hatchlings were released when umbilical scars were closed. Prior to release hatchlings were examined for any external sign of disease (i.e., fungus, botfly larvae) and weighed. Although all hatchlings were examined, due to logistical difficulties it was not always possible to weigh hatchlings prior to release. We were able to weigh 409 (68%) of the 599 released hatchlings (145 and 264 from Araguari and Falsino rivers, respectively). Healthy hatchlings were then released close to the original nesting area. By the time of hatchling release (up to 1 month after rescue) all nesting areas had been flooded and were totally submerged. We therefore selected release areas that were upstream and close (within 200 m) to the original nesting areas with relatively calm water and access to shelter and food along the river banks (i.e., small perennial streams that join the main rivers).

## Data analysis

All statistical analysis and graphics production were undertaken within the R language and environment for statistical computing (*R Development Core Team, 2020*). One-way Analysis of Variance (ANOVA) was used to test for differences in body mass and condition between hatchlings from Araguari and Falsino rivers. We calculated two body condition indexes for released hatchlings to evaluate the relative condition on the day they were released. The indexes were obtained from the residuals of Generalized Additive Models (GAMs, *Wood, 2011*, *2017*) of the response of ln transformed body mass predicted separately by ln transformed straight-line carapace length and ln transformed carapace area. The carapace area (cm$^2$) was calculated as an ellipse by standard formula from the radius of the major (*ra*) and minor (*rb*) axis (expressed in cm) as follows:

Carapace area $(\text{cm}^2) = \pi \times ra \times rb$

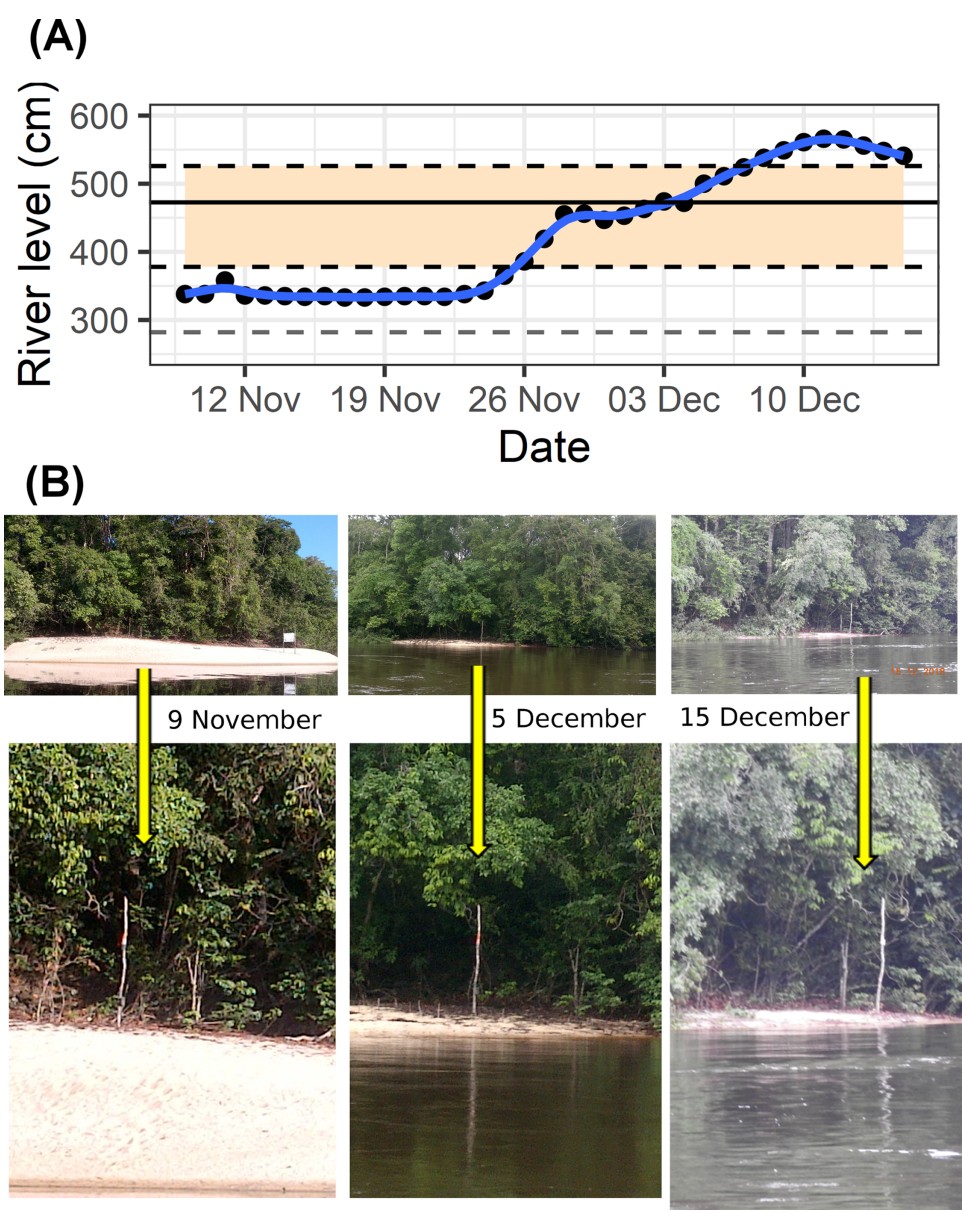

**Figure 2 Rapid river level rise.** Illustrating increase in river water level between 9 November and 15 December 2018. (A) Daily river level values (black points) and blue line illustrating trend. Horizontal shaded area spans levels at which 74 *Podocnemis unifilis* nests were submerged, showing levels corresponding to minimum and maximum nest heights (black dashed lines) and mean nest height (solid horizontal black line). Horizontal gray dashed line shows *Y*-axis reference river level (the minimum value (282 cm) recorded since records started (22 March 1981)). (B) Riverside photos of one of the highest nesting areas showing changes in river level. Yellow arrows indicate location of the same tree trunk as a reference point.

We used GAMs to avoid known problems of residual body condition indexes obtained from linear (ordinary least square) regression, namely the assumption that mass and body size relationships are linear (*Green, 2001*). GAMs include non-parametric smooth terms and are able to model non-linear relationships (*Wood, 2011*) such as those

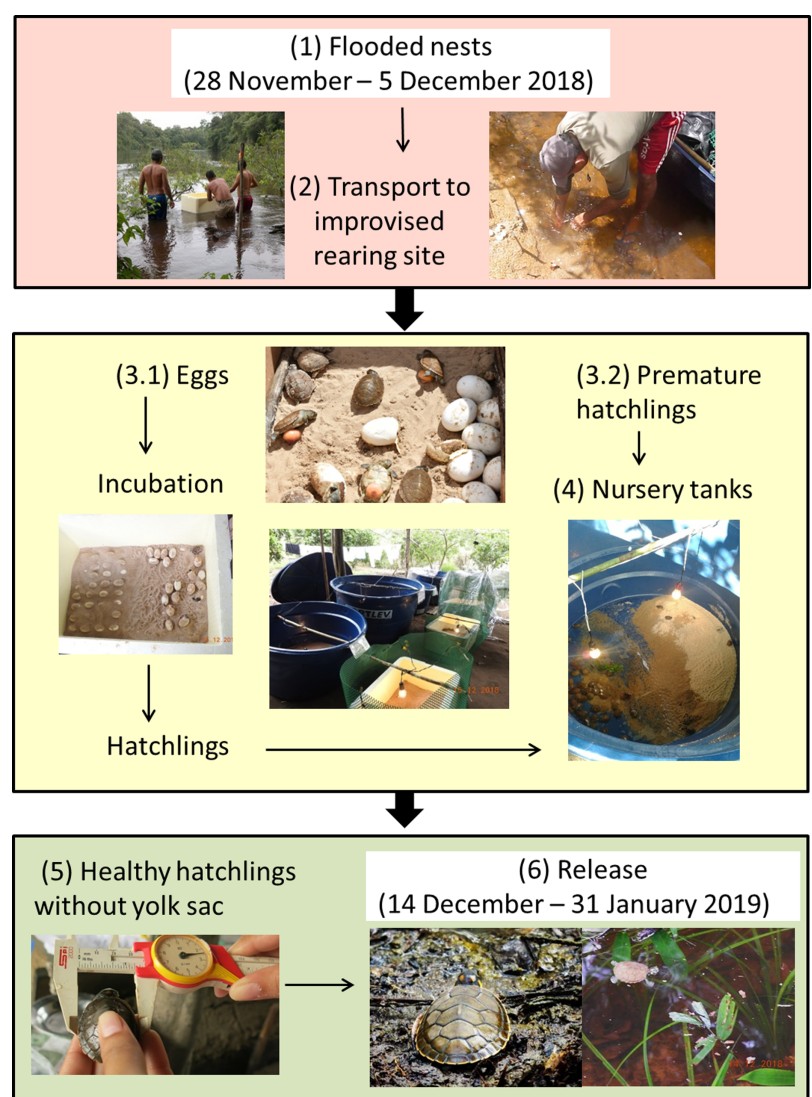

**Figure 3 Community-based rearing.** Illustrated schematic showing the steps taken to rear *Podocnemis unifilis* eggs and hatchlings submerged by rapid river level rise along 75 km of rivers in the eastern Brazilian Amazon.

expected between mass and body size (length or area) as body size changes and growth (a multiplicative process) occurs (*Green, 2001*; *Peig & Green, 2010*).

## RESULTS

The river water level rose 2.2 m (from 343 cm to 561 cm) over 16 days between 24 November and 10 December 2018 (Fig. 2). On average *P. unifilis* nests were laid at a height equivalent to the river level of 472 cm (Fig. 2). The highest nests were laid at 526 cm, which means that 100% of nests would have been entirely submerged by 8 December (Fig. 2). Eggs and premature hatchlings were recovered from a total of 74 submerged nests in 16 nesting areas along 74.6 km of rivers (Table 1). On average nests were laid at similar heights above the water along both rivers (Table 1), but submerged nests were detected

**Table 1 Characteristics of saved nests along the Araguari river basin.** Summary of *Podocnemis unifilis* nesting areas submersed due to flooding along two rivers.

| River | km | Count (areas/nests) | | Saved nest characteristics | | |
| --- | --- | --- | --- | --- | --- | --- |
| | | [a]Total | Saved | [b]Height above water (m) | Age (days) | Submersion time (days) |
| Araguari | 46.4 | 5/37 | 5/33 | 0.75 (0.05–1.10) | 39.1 (35–70) | 0.9 (0–2) |
| Falsino | 28.2 | 24/123 | 11/41 | 1.30 (0.35–1.79) | 45.7 (38–72) | 1.3 (0–2) |
| Overall | 74.6 | 29/160 | 16/74 | 0.91 (0.05–1.79) | 42.4 (35–72) | 1.1 (0–2) |

Notes:
[a] Total of monitored nesting areas with nests at the time of flooding. This total excludes nests recorded as predated, harvested or hatched prior to water level rising (26 November 2018).
[b] Height above river water level at time of nesting, mean values with minimum and maximum in parentheses.

**Table 2 Egg and hatchling survival.** Survival of eggs and premature hatchlings recovered from 74 submersed *Podocnemis unifilis* nests along the Araguari river basin.

| River | Eggs saved | Egg survival | Hatchlings saved | Hatchling survival | Hatchlings released | |
| --- | --- | --- | --- | --- | --- | --- |
| | | | | | Count | Weight[a] (g) |
| Araguari | 388 | 59.0% (229/388) | 11 | 94.6% (227/240) | 227 | 17.6 (9.8–23.2) |
| Falsino | 538 | 63.6% (342/538) | 54 | 93.9% (372/396) | 372 | 18.2 (11.1–21.6) |
| Total | 926 | 61.7% (571/926) | 65 | 94.2% (599/636) | 599 | 18.0 (9.8–23.2) |

Note:
[a] Weight values from 409 (68%) of the 599 released hatchlings (145 and 264 from Araguari and Falsino rivers, respectively).

and recovered earlier along the Araguari river (between 30 November and 2 December) compared with the Falsino river (between 1 and 5 December), which had the highest nesting areas with nests laid 69 cm above the maximum height on the Araguari river (Table 1).

Nests were submerged for 0 to two days (Table 1). Premature hatchlings were only recovered alive from waterlogged nests that had water infiltrating from below (i.e., 0 days submerged). Eggs were recovered from nests that were waterlogged (0 days submerged) to two days submerged. At the time of recovery the depth of water ranged between 0.05 to 1.30 m above the nests. The overall survival rate of eggs and pre-mature hatchlings was 60.4%, with 599 hatchlings released from 991 eggs and pre-mature hatchlings (Table 2). A total of 582 hatchlings emerged from 926 eggs that had been submerged for up to 2 days. Hatchling emerged more than a month after submersion, with the last hatchlings emerging on 10 January 2019 (36 to 40 days after recovery). The vast majority (94.2%) of emerged and pre-mature hatchlings survived (Table 2) and all surviving hatchlings were released by 31 January 2019.

The body mass of released hatchlings varied two fold (from 9.8 to 23.2 g, Table 2). Although mean body mass was significantly greater for hatchlings released on the Falsino river (Table 2, one-way ANOVA, $F_{1,407} = 8.00$, $P = 0.005$), the increase of 0.6 g compared with body mass of Araguari hatchlings represented only 3% of the mean body mass (18.0 g) from all released hatchlings (Table 2). The relationships between ln transformed body mass and ln transformed body size (carapace length and area) were not linear (Fig. 4, GAM effective degrees of freedom (EDF) = 5.5 and 7.1 for straight-line carapace length

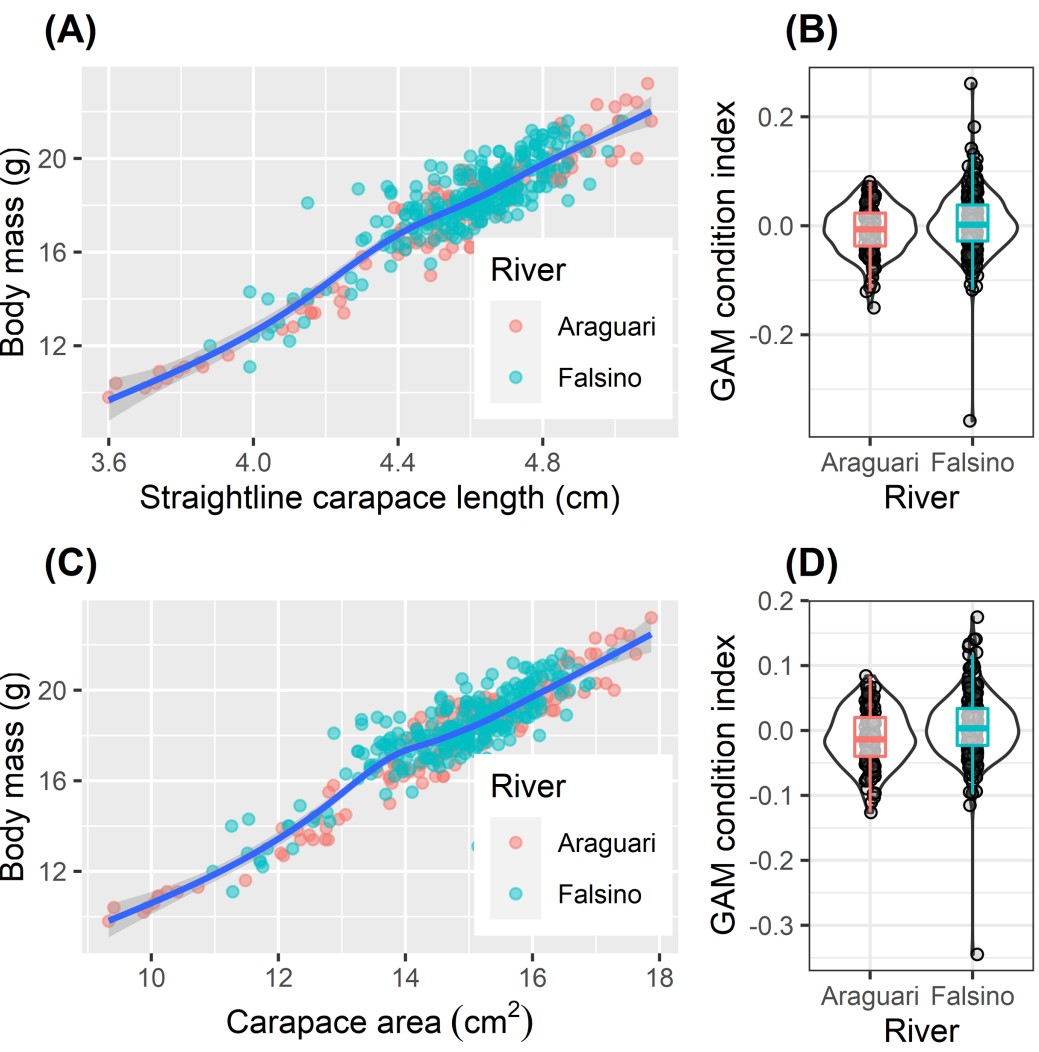

**Figure 4 Hatchling condition.** Body size relationships (A and C) and residual body condition indexes (B and D) for *Podocnemis unifilis* hatchlings released after nest submersion. Body size relationships (A and C) show untransformed values with GAM model trend line to aid visual interpretation. Shaded points show values of from Araguari (red) and Falsino (blue) river hatchlings. Residual body condition indexes (B and D) compared between hatchlings from Araguari and Falsino rivers. GAM condition index is the residuals obtained from modeling the response of ln transformed body mass predicted by two body size indicators (B) ln straight-line carapace length (GAM $R^2_{adj}$ = 0.864, Deviance explained = 86.6%) and (D) ln carapace area (GAM $R^2_{adj}$ = 0.878, Deviance explained = 88.0%). Violin-plots show density distribution of values, and colored boxplots show median values and the lower and upper hinges corresponding to the first and third quartiles (25th and 75th percentiles).

and carapace area respectively). The body condition index values ranged from −0.15 to 0.08 and −0.36 to 0.26 from the straight-line carapace length and carapace area models respectively (Fig. 4). Hatchling body condition increased slightly in Falsino hatchlings (Fig. 4). Body condition based on ln straight-line carapace length differed weakly between rivers (one-way ANOVA, $F_{1,407}$ = 5.51, $P$ = 0.019) and a similar pattern was found with the residual body condition index derived from ln carapace area (one-way ANOVA, $F_{1,407}$ = 11.47, $P$ = 0.0008). Although statistically significant, there were only small

differences in mean condition index values between rivers (Araguari (−0.008, −0.011) and Falsino (0.004, 0.006), mean values from straight-line carapace length and carapace area models respectively) and the distributions of body condition values were also similar (Fig. 4). Additionally, the interquartile range of body condition index values for hatchlings from both rivers overlapped zero (Fig. 4).

## DISCUSSION

To our knowledge we demonstrate for the first time that *P. unifilis* embryos are resistant to short-term submersion. We also show that healthy hatchlings survived and were successfully released following egg submersion. Early stages of freshwater turtles are known to suffer elevated mortality (*Iverson, 1991*). The mortality of *P. unifilis* eggs and hatchlings is usually elevated and varies widely, typically with 60–100% mortality in natural settings (*Caputo, Canestrelli & Boitani, 2005*; *Escalona & Fa, 1998*; *Ferreira Júnior & Castro, 2010*). Additionally, previous studies show that human harvest is also a major cause of nest loss in our area (*Norris, Michalski & Gibbs, 2018b*; *Quintana et al., 2019*). As such the release of hatchlings from over half of the submerged eggs (that would otherwise have suffered 100% loss) is within the expected survival of natural nests.

The average body mass (18.0 g) of released hatchlings was similar to values reported elsewhere for this species. Previous studies report mean hatchling weights of 17.8 to 20.0 g in Colombia (*Páez & Bock, 2004*), 17.3 g in Ecuador (*Caputo, Canestrelli & Boitani, 2005*) and 20.8 g from hatchlings (*n* = 99) along the Falsino River in 2011 (*Arraes, 2012*). Although we obtained statistical differences in body mass between hatchlings from Araguari and Falsino rivers, such a small difference (0.6 g) is unlikely to be biologically relevant. We were unable to monitor the long-term survival of the released hatchlings; however, there is no reason to expect that they would experience any negative impacts of short-term rearing. We believe that we effectively reduced known issues of captive rearing (*Burke, 2015*) by maintaining hatchlings at low densities and for a short period of time.

Different eggshell types may provide different levels of protection to embryos (*Deeming, 2018*). The rigid cased *P. unifilis* eggshells may help to explain the resistance of embryos to short-term submersion (*Packard, Packard & Boardman, 1982*; *Winkler, 2006*). Although *P. unifilis* eggs are rigid, the shells remain somewhat permeable, for example allowing passage of agro-toxins (*Hirano et al., 2019*). Yet *P. unifilis* eggshells have a low pore density (*Winkler, 2006*; *Winkler & Sánchez-Villagra, 2006*), which is likely to limit exchange of water with the environment (*Packard, Packard & Boardman, 1982*).

Although rigid eggshells can resist hydric changes, embryos are susceptible to changes in temperature associated with submersion. As *P. unifilis* only occurs within the tropics (*Norris et al., 2019*; *Rachmansah, Norris & Gibbs, 2020*), temperature is likely to be a key determinant for the survival of all stages and it is unlikely that embryos have any adaptation to resist low temperatures such as those experienced under flowing rivers. The temperature range for successful incubation of *P. unifilis* nests is typically between 27–32 °C (*De Souza & Vogt, 1994*; *Ferreira Júnior & Castro, 2006*; *Páez & Bock, 1998*; *Páez & Bock, 2004*). Although temperature dependent sex determination has been well studied in *P. unifilis* (*De Souza & Vogt, 1994*), other aspects such as embryo survival in

relation to temperature remain poorly investigated. For example, in the pivotal study for the species, *De Souza & Vogt (1994)* state "there was high mortality among the eggs switched from 34 C to 28 C", providing no information on when the change was made or quantity (*n* or %) that died.

Intervening to prevent losses is often a necessary component of wildlife conservation projects. Yet, there are ethical and moral concerns associated with these actions. Turtles can be considered a special case in several ways. All turtles are oviparous and *P. unifilis* do not provide any post-oviposition parental care, therefore, no surrogates for parental care and training are needed to rear eggs and hatchlings (*Burke, 2015*). Indeed rearing *P. unifilis* hatchlings has become an integral part of management approaches across the species range (*Harju, Sirén & Salo, 2018*; *Páez et al., 2015*; *Sinovas et al., 2017*). We took reasonable precautions to address ethical concerns, including providing appropriate temperature, water and food (reflected in the increase in body weight), avoiding risks of disease spread from exotic species, and ensuring that premature hatchlings were retained for the minimal time necessary before release to the wild. We suggest that the improvised response to unprecedented river level rise was appropriate and provided a positive contribution to the conservation of *P. unifilis* populations along the rivers. Continued long-term monitoring is required to establish if such interventions enable freshwater turtle populations to persist and even recover across Amazonian rivers increasingly impacted by anthropogenic harvests (*Norris et al., 2019*) and flow-rate alterations (*Castello et al., 2013*).

Considering the drastic changes to Amazonian river flow regimes it seems reasonable for management actions to include contingency plans for recovery and release of submerged *P. unifilis* eggs and hatchlings. A total of US$5007 was spent (exchange rate of 3.78 Brazilian R$ to 1 US$, food, gasoline, daily stipends, rearing equipment) during the two month improvised rearing process (28 November 2018–31 January 2019, Fig. 3), representing an overall financial investment of US$8.36 for each *P. unifilis* hatchling released. In comparison, the community based monitoring implemented with two teams required on average US$2302 per month (food, gasoline, daily stipends) that is, US$4604 over a two month period. Therefore improvised rearing does not necessarily represent an excessive financial investment beyond that expected as part of the actions for conservation and monitoring of the species.

## CONCLUSION

The resistance of *P. unifilis* eggs and embryos to short-term submersion could help explain their widespread distribution across tropical South America. The continued decline of the species across its range shows that such adaptations are insufficient to respond to unprecedented 21st century changes across riverine habitats. Our successful rescue of eggs and premature hatchlings from flooded nests and subsequent release of healthy hatchlings demonstrates the viability of recovery actions. Such actions are likely to be increasingly necessary across rivers where not only are flow rates affected by damming but also anthropogenic harvests can drive unsustainable harvests of turtle nests. We suggest that such recovery actions should be included in action plans for species conservation and

recovery. Additionally, hydroelectric developments should evaluate localized changes in flow rates to avoid seasonal flooding of nesting areas and pro-actively avoid the need for recovery actions.

## ACKNOWLEDGEMENTS

The Instituto Chico Mendes de Conservação da Biodiversidade (ICMBIO) and the Universidade Federal do Amapá provided logistical support. We are grateful to Manuel Justino de Abreu, Fabio Cardoso de Abreu, Cremilson and Cledinando Alves Marques, Raimundo Marques, Alvino Pantoja Leal, Arlete Pantoja Leal and Gilberto Souza for their invaluable assistance during fieldwork. We also thank the volunteer interns who aided in nest area monitoring and egg and hatchling rearing. This study is dedicated to the memory of Elma Palleta Normandia Marques.

### Funding

Funding was provided by the United States National Academy of Sciences and the United States Agency for International Development through the Partnership for Enhanced in Research (http://sites.nationalacademis.org/pga/peer/index.htm), award number AID-OAA-A11-00012 to Darren Norris, James P. Gibbs and Fernanda Michalski. Fernanda Michalski received a productivity scholarship from the Brazilian National Council for Scientific and Technological Development (CNPq - process 301562/2015-6 and 302806/2018-0) and is funded by CNPq (Process 403679/2016-8). The funders had no role in study design, data collection and analysis, decision to publish, or preparation of the manuscript.

### Grant Disclosures

The following grant information was disclosed by the authors:
United States National Academy of Sciences and the United States Agency for International Development: AID-OAA-A11-00012.
Brazilian National Council for Scientific and Technological Development (CNPq): 301562/2015-6 and 302806/2018-0.
CNPq: 403679/2016-8.

### Competing Interests

The authors declare that they have no competing interests.

### Author Contributions

- Darren Norris conceived and designed the experiments, performed the experiments, analyzed the data, prepared figures and/or tables, authored or reviewed drafts of the paper, and approved the final draft.
- Fernanda Michalski conceived and designed the experiments, performed the experiments, prepared figures and/or tables, authored or reviewed drafts of the paper, and approved the final draft.

- James P. Gibbs conceived and designed the experiments, authored or reviewed drafts of the paper, and approved the final draft.

## Animal Ethics

The following information was supplied relating to ethical approvals (i.e., approving body and any reference numbers):

The Animal Use Ethics Committee from the Federal University of Amapá (UNIFAP) approved the study (CEUA-UNIFAP approval 009/2017).

## Field Study Permissions

The following information was supplied relating to field study approvals (i.e., approving body and any reference numbers):

Brazilian Ministério do Meio Ambiente ("MMA") provided permit numbers SISBIO 63668-1 and 63668-2. IBAMA provided authorization to conduct research in FLONA (IBAMA/SISBIO permit 49632-1).

## Data Availability

The raw measurements used in analysis and figures are available in the Supplemental Files.

## Supplemental Information

Supplemental information for this article can be found online at http://dx.doi.org/10.7717/peerj.9921#supplemental-information.

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
