# Peer review of "Community based actions save Yellow-spotted river turtle (Podocnemis unifilis) eggs and hatchlings flooded by rapid river level rises"

_PeerJ, doi:10.7717/peerj.9921_

## Round 0.1 · original submission · Minor Revisions

Your manuscript has now been assessed by three reviewers who all found your work to be well-conducted and well-written. The reviewers have outlined a series of edits and revisions that would improve this manuscript. However, most of these are relatively minor to address.

Pay special attention to Reviewer 1's comments about additional methods details (e.g., how were nests located) as well as what these data and rising rivers mean for population status. Can the authors perhaps incorporate a PVA? Additionally, Reviewer 3 provides important comments about further detailing who does the monitoring and what existing management looks like.

A couple of additional minor comments:

Line 30-31 there are a few words missing in the sentence about egg and hatchling survival.

For Figure 2A, please change the X-axis from Portuguese abbreviations to English.

·

Basic reporting

The authors discuss the emergency salvage of P. unifilis eggs and hatchlings in response to high river levels near nesting sites. Their results indicate that eggs can persist for at least two days after being submerged by water and that community intervention allowed a similar number of hatchlings to be released as would have likely occurred during a normal year. The authors therefore suggest that this type of management strategy could be an effective tool for P. unifilis conservation in years where river flooding is extensive during the nesting season.

The article is clearly written and provides sufficient literature and background information. It meets all of the basic reporting requirements and the data are provided in sufficient detail. However, it was not clear to me whether the raw data would be available as a supplementary file or uploaded to an online repository.

I have noted several places where the text could be improved below.

Experimental design

Lines 110–114: This section would benefit from some additional detail. How were nests located on searches? Were nesting areas searched in some sort of grid? What does at a standardized speed mean?

Validity of the findings

The author’s problem statement (lines 75–77) and conclusions (line 270) both mention the effects of flooding on P. unifilis populations. However, the data presented in the manuscript offer little insight into what is actually occurring with these populations. As presented the results indicate that the authors were able to release hatchlings in a single year at a rate similar to natural hatching rates. Without additional data on populations and hatchling survival there is no way to assess what effect this will have on populations, especially given that this species can experience 100% nest mortality (line 245). Furthermore, there is almost no discussion of how the damming of the rivers has practically impacted water levels at these sites. Is there any data to indicate that flooding threatening nest survival is becoming more common in the study area (the methods seem to indicate this was an extreme event)? Thus, I think that in both places indicated above the author’s should reword their statements to focus on nest and hatchling survival and not the P. unifilis population as a whole.

Results: Recommend adding effect sizes to lines 209 and 114, especially with indication that increases were marginal and weak.

Discussion: The discussion should be rearranged to focus on the main result first: that eggs and hatchlings were successfully released at similar mortality rates and weights to those observed in natural conditions after they were submerged by high water levels. The two paragraphs discussing the biological reasons why eggs could survive inundation should be moved to later in the discussion. As indicated above, I also think the discussion would benefit from another paragraph that focuses on flooding along these rivers, the causes, and if it is getting worse over time.

Additional comments

Minor Comments

Line 29: Recommend moving this sentence to Line 27.
Line 32: Add scientific name of species, also recommend adding a mention of the study species earlier in the Abstract (e.g., line 24).
Line 67: Add apostrophe to species.
Line 69: Add comma after documented.
Line 82: Delete “into consideration”.
Line 88: Change was to were.
Lines 94–95: Recommend deleting “and here … nesting areas”.
Line 105: Add comma after community.
Line 125: Delete “Differently to … by our team,”.
Line 128: Reword or delete “with the aim … research team”.
Line 147–148: Add parentheses around i.e., statement.
Line 155: Replace each with every.
Line 157: Add comma after closed.
Line 161: Add comma after rivers.
Lines 173–174: Delete “on the Araguari and Falsino rivers”.
Line 191: Replace be with have been.
Line 194: Remove Table 1 reference.
Line 198: This paragraph needs a better topic sentence. Maybe something similar to lines 202–203.
Line 200–201: This sentence is repetitive with the first sentence in the paragraph and could probably be removed.
Line 208: Remove Table 2 reference.
Lines 221–222: Remove second sentence and combine first sentence with another paragraph.
Line 243: Replace early stages with “egg and hatchling”.
Line 260: Reword to: “Yet, there are ethical and moral concerns associated with these actions.”
Line 265–268: Reword. Currently not a complete sentence.

Table 1: Reporting the number of nesting areas seem to add unnecessary complexity to this table. Recommend just reporting the number of nests in the ‘Total’ and ‘Saved’ columns.
Table 2: This table should indicate somewhere that the weight measurements were only collected from 409 hatchlings at release.
Figure 2: Change dez to dec.

Reviewer 2 ·

Basic reporting

No comment

Experimental design

No comment

Validity of the findings

No comment

·

Basic reporting

1. Line 276: “the normal community based monitoring” should be changed to the “previously” or “currently” implemented monitoring. Clearer context for this monitoring can be provided in line 105, as stated below in the comments for Experimental Design.
2. Lines 76-77: providing brief example(s) of “the known impacts” of flooding due to hydroelectric damming can help provide the reader more context.
3. End of statement in line 53 should be cited.
4. Professional English is used throughout the manuscript. There are, however, instances where "yet" can be replaced for clearer transitions between sentences (e.g. with "however", "moreover", etc.). For example, see use of the term and transition between sentences in lines 66, 160, and 228.
5. In line 72, “climate changes” should be changed to “climate change”.
6. Structure of the sentence in lines 120-124 should be modified or edited for clarity and better flow.
7. Sentence in lines 126-129 should be edited for clarity, particularly to clarify the statement “had been moved with the aim of the local community…”
8. Lines 162-163: The language of the sentence can be improved. For example, to: “Healthy hatchlings were then released along the river to the original nesting area.”
9. Line 248: “expect” should be changed to “expected”.
10. Line 250: “report mean” should be changed to “report a mean”.
11. Line 276: “sac” should be changed to “sacs”.
12. Please edit the sentence in lines 265-268 for clarity and flow. As is now, it reads as an incomplete statement, perhaps due to the transition “and that premature hatchlings…”

Experimental design

1. More context should be provided regarding the monitoring that has occurred in the study area since 2011. For example, who does the monitoring? To what extent are local communities involved and is co-management or support provided by external agencies? The authors can choose to expand on this in line 105, where the collaborative monitoring scheme is introduced.
2. Lines 97-98: Sentence “The study was conducted…(Fig. 1).” should be placed as the first sentence of the paragraph describing the study area.

Validity of the findings

No comment.

Additional comments

Overall, the manuscript is technically sound and conclusions are well stated. As you will see in the suggested comments, more context should be provided regarding the current management schemes.

---

## Round 0.2 · Minor Revisions

Thank you for your thorough and thoughtful revisions and responses to the reviewers. The reviewer has only a small number of very minor edits that will improve this manuscript. I look forward to seeing the next version of this piece.

·

Basic reporting

no comment

Experimental design

no comment

Validity of the findings

no comment

Additional comments

The authors have done a good job addressing the comments provided by myself and the other reviewer. I had just a handful of other suggestions to improve clarity.

Line 230: Delete ‘turtle’.
Line 230-237: I suggest adding one additional sentence after the 1st sentence to indicate that hatchlings were then released from these flooded nests. The last sentence could also be strengthened by indicating that mortality would have been 100% without the intervention.
Lines 244-245: Hyphenate ‘long term’ and ‘short term.’
Lines 273-277: Recommend rewording and combining these two sentences (e.g., We took reasonable precautions to address ethical concerns, including providing appropriate ….)

---

## Round 0.3 · accepted · Accept

I appreciate your continued thoughtful revisions based on the reviewers' comments and look forward to the publication of this manuscript.